# Are Intravenous Immunoglobulins Effective in Preventing Primary EBV Infection in Pediatric Kidney Transplant Recipients?

**DOI:** 10.3390/medicina61111967

**Published:** 2025-11-02

**Authors:** Nicola Bertazza Partigiani, Veronica Bertozzi, Maria Sangermano, Elisa Benetti

**Affiliations:** 1Pediatric Nephrology, Department of Women’s and Children’s Health, Padua University Hospital, 35128 Padua, Italybertozzi.veronica@gmail.com (V.B.); maria.sangermano@aopd.veneto.it (M.S.); 2Laboratory of Immunopathology and Molecular Biology of the Kidney, Pediatric Research Institute, Department of Women’s and Children’s Health, Padua University Hospital, 35128 Padua, Italy

**Keywords:** Epstein–Barr virus (EBV), pediatric kidney transplantation, post-transplant lymphoproliferative disorder (PTLD), intravenous immunoglobulins (IVIG), EBV prophylaxis

## Abstract

*Background and Objectives:* Primary Epstein–Barr virus (EBV) infection in pediatric kidney transplant recipients with donor/recipient mismatch (D+/R−) carries the highest risk of post-transplant lymphoproliferative disorder (PTLD). Current prophylactic strategies are not standardized. Intravenous immunoglobulins (IVIG), containing anti-EBV antibodies, have been proposed as a potential preventive option, but evidence is lacking. This single-center retrospective case–control study evaluated the efficacy of serial IVIG administration in preventing primary EBV infection and promoting long-term immunity in this high-risk population. *Materials and Methods:* We retrospectively analyzed 26 pediatric kidney transplant recipients (age 1–18 years) with EBV D+/R− mismatch and a median follow-up of 7.5 years. Fourteen patients received scheduled IVIG infusions (200 mg/kg monthly for six months post-transplantation), while twelve received no EBV-directed prophylaxis. The primary endpoint was the cumulative incidence of primary EBV infection, defined as EBV-DNA > 1000 copies/mL in peripheral blood. The secondary endpoint was Epstein–Barr Nuclear Antigen-Immunoglobulin G (EBNA-IgG) seroconversion. *Results:* Patients receiving IVIG were significantly younger than controls (median age 4.2 vs. 10.8 years, *p* = 0.01). No significant variations were observed between groups in renal function or immunosuppressive levels during follow-up. IVIG prophylaxis was unexpectedly linked to a higher cumulative incidence of EBV infection compared with controls (64% vs. 25%, *p* = 0.047). Time-to-event analysis confirmed an increased, although not statistically significant, risk of EBV acquisition in the IVIG group (Hazard Ratio [HR] 3.24, 95% Confidence Interval [CI] 0.87–12.01; *p* = 0.079). EBV-specific immunity, assessed by EBNA-IgG seroconversion, was comparable between groups (HR 1.78; *p* = 0.45), confirming no immunological advantage of IVIG. One IVIG-treated patient (7.1%) developed PTLD, while none did in the control group. *Conclusions:* Scheduled IVIG administration during the first six months after transplantation does not constitute an effective strategy to prevent primary EBV infection or to enhance long-term immunity in high-risk EBV D+/R− pediatric kidney recipients and may even increase susceptibility to viral acquisition. These findings argue against the use of IVIG as EBV prophylaxis in this population.

## 1. Introduction

Epstein–Barr virus (EBV) is a herpesvirus that establishes lifelong latency in memory B lymphocytes. In immunocompetent individuals, primary EBV infection is usually asymptomatic or presents as infectious mononucleosis [1]. In immunocompromised individuals, such as pediatric kidney transplant recipients, EBV infection can cause uncontrolled B-cell growth and post-transplant lymphoproliferative disorder (PTLD), a serious complication of organ transplantation [2]. PTLD encompasses a spectrum of EBV-related lymphoproliferative disorders caused by impaired T-cell immune surveillance due to immunosuppressive therapy. The disease ranges from early, polyclonal B-cell hyperplasia to malignant aggressive monoclonal lymphoma. The current standard management includes stepwise reduction in immunosuppression, which may be sufficient in early lesions, and the use of anti-CD20 monoclonal antibody therapy (rituximab) for persistent viremia or established disease; chemotherapy is reserved for refractory or monomorphic PTLD [3]. The incidence of PTLD is highest in EBV-naïve recipients who acquire a primary EBV infection post-transplant from an EBV-seropositive donor (D+/R− mismatch). Pediatric recipients, particularly the youngest, are highly vulnerable, as up to 40% of children under five years of age are EBV-seronegative at the time of transplantation [4,5]. Despite the strong association between EBV infection and PTLD, no standardized strategy has been established to prevent primary EBV infection in high-risk pediatric kidney recipients. In current clinical practice, EBV viral load is monitored post-transplant by PCR, and immunosuppression is pre-emptively reduced when EBV-DNA levels increase [3,6]. While this approach is effective in some cases, it carries the inherent risk of acute rejection, and it may fail to control EBV-driven proliferation in EBV-naïve hosts [6].

Intravenous immunoglobulins (IVIG) have been proposed as a potential prophylactic strategy [7]. IVIG contains polyclonal IgG, including antibodies directed against EBV viral antigens, and is commonly used for immunomodulation or passive immunity against various pathogens. The proposed mechanisms for IVIG effectiveness against EBV include neutralization of viral particles, enhancement of phagocytosis, and modulation of immune responses.

Thus, evidence for IVIG as EBV prophylaxis after solid organ transplantation is limited and conflicting, as early reports suggested a potential benefit, but in a pediatric liver transplant cohort treated with CMV-IVIG did not demonstrate a significant reduction in EBV disease or PTLD, and contemporary guidelines do not endorse routine IVIG for EBV prevention [7,8]. Notably, no controlled studies have addressed pediatric kidney recipients, highlighting the unmet need our study seeks to address.

The study was based on the hypothesis that IVIG administration, by providing passive EBV-specific antibodies, might reduce the risk of primary EBV infection and facilitate the development of durable immune protection. Accordingly, the objective of this retrospective study was to assess whether scheduled IVIG infusions during the first six months after transplantation were associated with a reduced incidence of primary EBV infection and improved long-term EBV-specific immunity in high-risk pediatric kidney transplant recipients with an EBV D+/R− mismatch.

## 2. Materials and Methods

### 2.1. Study Design and Subjects

This single-center case–control study included a cohort of 14 pediatric kidney transplant recipients (aged 1–18 years) who underwent transplantation at our Center between January 2015 and March 2018, presented with EBV D+/R− mismatch at the time of transplantation, and received IVIG as EBV prophylaxis. The control group included an historical cohort of 12 EBV D+/R− pediatric patients transplanted between January 2013 and December 2014 who did not receive any EBV prophylaxis.

Inclusion criteria were: (i) pediatric kidney transplant recipients aged between 1 and 18 years at the time of transplantation; (ii) documented EBV D+/R− mismatch; (iii) follow-up of at least 24 months post-transplantation; and (iv) availability of complete clinical, virological, and immunological data in the first 24 months. Exclusion criteria were: (i) prior EBV seropositivity before transplantation; (ii) multi-organ transplantation; (iii) follow-up shorter than 24 months after transplantation; or (iv) incomplete post-transplant EBV monitoring.

All included patients reached the 24-month follow-up timepoint. Three patients were subsequently transferred to other centers after the second post-transplant year and were therefore censored from further observation, while the remaining participants completed at least five years of follow-up.

For each patient, recorded data included: age at transplantation, sex, number of transplants, primary kidney disease, dialysis status at transplantation, donor type (living or deceased), Human Leucocyte Antigens (HLA) mismatch, induction therapy (basiliximab or thymoglobulin), maintenance therapy (tacrolimus, cyclosporine, prednisone, mycophenolate mofetil, or everolimus), use of rituximab for chronic EBV infection, and occurrence of PTLD. EBV-DNA and antibody profiles (Viral Capsid Antigen [VCA], Early Antigen [EA], EBNA), creatinine, estimated glomerular filtration rate (eGFR) sec Bedside-Schwartz, urea, uric acid, therapeutic drug monitoring of blood levels, pre-dose tacrolimus and cyclosporine 2 h post-dose (C2) were regularly monitored and assessed at 6-, 12-, 24-, and 60 months post-transplantation, as well as at the last available follow-up [9].

According to our protocol, immunosuppression was induced with basiliximab (10 mg for patients < 30 kg and 20 mg for those > 30 kg, administered on days 0 and 4 post-transplantation) for first transplants, or with anti-thymocyte globulin (ATG) for second transplants. Maintenance immunosuppression included a calcineurin inhibitor (CNI, either tacrolimus or cyclosporine), mycophenolate mofetil or everolimus, and prednisone.

Transplant recipients and their families must follow standard post-transplant infection-prevention measures, avoiding school attendance for the first three months and avoiding crowded environments and contact with potentially infected or symptomatic individuals during the first six months after transplantation.

All patients underwent protocol biopsies at 6, 12, and 24 months post-transplant, regardless of graft function. Additional biopsies were performed in cases of an unexplained increase in serum creatinine exceeding 20% from baseline or in the presence of persistent proteinuria, defined as a urinary protein-to-creatinine ratio (UPCR) > 0.2 mg/mg. All biopsy specimens were reviewed by experienced renal pathologists and classified according to the 2017 Banff criteria [10].

Episodes of biopsy-proven acute cellular rejection were treated with pulse intravenous methylprednisolone. Patients who developed late antibody-mediated rejection (AMR) followed a standardized protocol including plasmapheresis, intravenous immunoglobulin (IVIG), and anti-CD20 monoclonal antibody therapy, while those with cellular rejection received three doses of intravenous methylprednisolone.

Antiviral prophylaxis with valganciclovir was administered only to Cytomegalovirus (CMV)-negative recipients receiving the graft from CMV-positive donors (D+/R−), at standard doses [11,12].

The procedural timeline with IVIG infusion schedule and EBV testing is presented in Figure 1. Testing for IgM antibodies against EBV-VCA, IgG antibodies against EBV-EA, -VCA and -EBNA) was performed by Enzyme-Linked Immunosorbent Assay (ELISA) (DiaSorin; Saluggia, Italy) at baseline, months 6, 12, 24 and 60. Seropositivity for EBV was defined as positivity for anti-EBNA IgG, while an isolated positivity for anti-VCA IgG was not considered seroprotective. EBV DNA was quantified by real-time PCR on the day of transplantation (baseline), then weekly until day 30, every two weeks until month 3, monthly until month 12, and every 3 months thereafter, as previously described [13].

EBV D+/R− children received IVIG according to the following schedule: 200 mg/kg per dose on day 0 (day of transplantation) and on days 1, 7, 14, and 21 post-transplantation. Thereafter, the same dosage was given every 3 weeks for 3 months, and subsequently once monthly until the sixth month post-transplantation. In cases of a progressive increase in EBV load (>4000 copies/mL) in anti-EBNA IgG–negative children, immunosuppression was reduced by discontinuation of mycophenolate, followed by decreasing calcineurin inhibitor exposure by 30–50% [14].

In selected cases with very high EBV load (>100,000 copies/mL) persisting despite immunosuppression tapering and/or lasting for more than 3 months, pre-emptive anti-CD20 therapy with rituximab at a dose of 375 mg/m^2^ was administered [6]. Patients treated with rituximab underwent a standardized pre-infusion assessment to exclude latent infections and evaluate immune status. The work-up included microbiological and serological screening for viral infections, Quantiferon-TB testing, serum immunoglobulin quantification, and lymphocyte subset analysis with CD20 measurement. Chest X-ray and abdominal ultrasound were also performed to exclude PTLD. Complete blood count and lymphocyte subsets were reassessed 7–10 days after infusion to monitor hematologic response. In cases with persistent CD20-positive B cells, a second rituximab infusion at a dose of 375 mg/m^2^ was administered.

All patients with EBV infection underwent imaging surveillance (abdominal and head–neck ultrasound) to exclude subclinical PTLD.

### 2.2. Endpoints

The primary endpoint of the study was to assess whether scheduled IVIG infusions administered during the first six months after transplantation in pediatric recipients with EBV D+/R− mismatch could represent an effective strategy to prevent primary EBV infection. This was evaluated by monitoring EBV-DNA detection at 6, 12, 24, and 60 months post-transplant, as well as at the last available follow-up. The time to first EBV-DNA detection in blood was recorded. A positive EBV-DNA result was defined as an EBV load > 1000 copies/mL with either persistence of viremia for at least three months in asymptomatic patients or a single positive sample in the presence of EBV-related symptoms.

As a secondary endpoint, we investigated whether scheduled IVIG infusions during the first six months after transplantation in EBV D+/R− mismatch pediatric recipients could promote the development of full immunological protection against the virus. This was assessed by EBNA-IgG seroconversion during follow-up at 12, 24, and 60 months post-transplant and at the last available follow-up. The 6-month timepoint was excluded from the survival analyses, as IVIG administration during this early period could significantly affect the results.

### 2.3. Statistical Analyses

Descriptive statistics were used to summarize baseline characteristics, immunological variables, and clinical outcomes. Continuous variables were reported as mean and standard deviation (SD) or median and interquartile range (IQR), depending on the distribution assessed using the Shapiro-Wilk test. Categorical variables were expressed as absolute frequencies and percentages. Comparisons between IVIG-treated and untreated patients were performed using Student’s *t*-test for normally distributed continuous variables and the Mann–Whitney *U* test for non-normally distributed data. Chi-squared or Fisher’s exact tests were used to compare proportions, as appropriate.

Time-to-event analyses were conducted using Kaplan–Meier estimates and Cox proportional hazards regression. Separate survival analyses were performed for the time to first EBV-DNA detection and for the time to EBV seroconversion, comparing the IVIG group with the control group. Median survival times, event-free survival probabilities at 6, 12, 24, and 60 months, and HR with 95% confidence interval (CI) were reported. Model performance was evaluated using the concordance index and likelihood ratio tests. Variables with a *p*-value < 0.20 at univariable Cox regression were entered into the multivariable model to account for potential confounding effects. Hazard ratios (HR) with 95% CI were calculated, and proportional hazards assumptions were verified. Model performance was evaluated using likelihood ratio, Wald, and score (log-rank) tests, as well as the concordance (C) statistic.

All analyses were conducted using Jamovi version 2.6.0 (Sydney, Australia; https://www.jamovi.org accessed on 17 October 2025).

## 3. Results

### 3.1. Population Characteristics

A total of 26 pediatric kidney transplant recipients were included in the study, of whom 14 received intravenous immunoglobulin (IVIG group) and 12 did not (control group). The mean age at transplantation was 6.92 years (SD 4.92); 65.4% were male, and 11.5% had previously undergone at least one kidney transplant. The median follow-up was 7.5 years (IQR 5.5–9.5).

Congenital anomalies of the kidney and urinary tract (CAKUT) were the leading cause of end-stage renal disease (61.5%), followed by ciliopathies (15.4%), glomerulopathies (15.4%), and neonatal asphyxia (7.7%).

Before transplantation, 57.7% of patients were on continuous peritoneal dialysis, 23.1% were pre-emptively transplanted, 15.4% had transitioned from hemodialysis to peritoneal dialysis, and 3.8% were on hemodialysis.

Most patients (34.6%) had four HLA mismatches, while the remainder had between two and five mismatches. Living donor transplantation was performed in 30.8% of cases.

Induction therapy consisted of basiliximab in 92.3% of patients and anti-thymocyte globulin in 7.7%; an additional 7.7% received both agents due to delayed graft function and/or calcineurin inhibitor toxicity. All patients received maintenance immunosuppression with corticosteroids, mycophenolate mofetil, and a CNI (tacrolimus and cyclosporine were each used in 50% of cases). Valganciclovir prophylaxis was administered in 73% of patients. In the immediate post-transplant period, 54% of children received at least one transfusion of red blood cells, platelets, or plasma, with a median of 2 transfusions (range 1–10).

Baseline characteristics by treatment group are summarized in Table 1. The median age at transplantation was significantly lower in the IVIG group compared to the control group (4.10 years [IQR 2.26–7.04] vs. 8.20 years [IQR 6.99–12.14]; *p* = 0.008). No statistically significant differences were observed between groups in dialysis modality prior to transplantation, number of HLA mismatches, donor type, primary kidney disease, or immunosuppressive regimen (all *p* > 0.05).

### 3.2. Kidney Function and Laboratory Test over Time

Renal function showed a gradual decline in eGFR values, paralleled by a progressive increase in serum urea concentrations (Figure 2). The Friedman test confirmed a significant change across follow-up visits for both parameters (eGFR χ^2^[3] = 19.2, *p* < 0.001; urea χ^2^[3] = 12.7, *p* = 0.005), indicating a progressive impairment of renal function within the study population during long-term follow-up.

Renal function and immunosuppressive parameters were evaluated longitudinally at 6, 12, 24, and 60 months post-transplant in both study groups (IVIG vs. control). As summarized in Table 2, median eGFR values showed a gradual decline over time in both groups, while urea levels exhibited a progressive increase. Conversely, serum uric acid, tacrolimus, and cyclosporine (C2) levels remained stable throughout the observation period, with no statistically significant differences between groups at any time point. No differences were observed between the IVIG and control groups. The incidence of rejection in the first 5 years after transplantation was comparable between the two groups (57% in IVIG group vs. 58% in the control group, *p* = 0.838).

### 3.3. EBV Viremia and Seroconversion After Transplantation

The proportion of EBV-DNA-positive patients in the IVIG and control groups at different time points is summarized in Table 2. In the IVIG group, EBV-DNA positivity was detected in 57% at 6–12 months, 50% at 24 months, and 28% at 60 months, compared to 25%, 25%, and 17% in the control group, respectively. Overall, the cumulative incidence of EBV-DNA positivity was significantly higher in the IVIG group than in controls (64% vs. 25%, *p* = 0.047).

Regarding the serological response, anti-EBNA IgG was detected at six months in 9 of 14 IVIG recipients (64%) and in 2 of 12 controls (17%, *p* = 0.012). At the 12-month visit, none of the IVIG-treated patients had a positive antibody test, whereas 2 of 12 controls (17%) tested positive. At 24 months, antibodies were present in 3 of 14 IVIG recipients (21%) and 3 of 12 controls (25%); at 60 months, in 5 of 14 IVIG patients (36%) and 2 of 12 controls (17%). Over the entire observation period, 43% of IVIG-treated patients and 25% of controls tested positive at least once (Table 3).

Only one patient in the IVIG group received rituximab 375 mg/m^2^ as a single dose 1 year after transplantation for EBV infection. One patient in the IVIG group developed PTLD, which required oncological assessment and chemotherapy, resulting in disease remission. However, no statistically significant differences were observed between the two groups in terms of PTLD occurrence (*p* = 0.331).

### 3.4. Time-to-Event Outcomes for EBV Viremia and Seroconversion in Children Receiving IVIG Prophylaxis After Kidney Transplantation

Time-to-event analysis for EBV-DNA detection showed no statistically significant difference between the two groups, with an HR of 3.24 (95% CI 0.87–12.01; *p* = 0.079) for IVIG-treated patients compared to controls (Figure 3). The estimated probability of remaining EBV-DNA negative at 60 months was 75% in the control group versus 35.7% in the IVIG group. Conversely, anti-EBNA IgG seroconversion rates were similar between the two groups; the Cox regression model yielded an HR of 1.78 (95% CI 0.44–7.08; *p* = 0.45) (Figure 4). The estimated proportions of patients who experienced EBV seroconversion at 60 months were 25% in the control group and 48% in the IVIG group.

At univariable analysis, variables showing an association with EBV DNA positivization at a *p*-value < 0.20 were entered into the multivariable Cox regression model (Table 4). Although no individual covariate reached statistical significance, the multivariable model demonstrated an overall acceptable fit (global likelihood ratio test *p* > 0.05) and moderate discriminative ability (C-index = 0.70). The estimated hazard ratios suggested a trend toward an increased risk of EBV-DNA positivization among patients with ciliopathies, and a possible protective effect of tacrolimus, although these associations did not achieve statistical significance. In contrast, the multivariable Cox regression model exploring predictors of EBV antibody positivization did not reach statistical significance (global likelihood ratio test *p* > 0.05), indicating that none of the included variables were independently associated with the development of serological positivity over time.

## 4. Discussion

The present study revealed a counterintuitive finding: intravenous immunoglobulin (IVIG) administration was associated with a higher incidence of EBV DNA positivization, rather than the expected protective effect. This observation challenges the presumed antiviral role of IVIG and suggests that its use in this setting may reflect indication bias, as younger or higher-risk patients were more likely to receive prophylactic IVIG. Chronic EBV infection after kidney transplantation represents a significant concern due to the associated risk of PTLD. Large registry data show that EBV seronegativity at the time of kidney transplantation confers more than a threefold increased risk of PTLD compared to seropositive recipients, as EBV-seronegative patients lack pre-existing virus-specific immunity and are therefore particularly vulnerable when receiving an organ from a seropositive donor (D+/R−). Major additional risk factors include intense T-cell–depleting immunosuppression and concurrent CMV infection, which may promote immune exhaustion and indicate impaired cellular immunity [2,15,16]. Primary EBV infection in pediatric kidney transplant recipients usually occurs following seroconversion due to de novo viral exposure or transmission from donor-derived lymphocytes contained within the graft. Because EBV is highly prevalent in the general population, with seropositivity rates exceeding 90% in young adults, donors are frequently EBV-seropositive [17]. However, matching donor and recipient pairs by EBV status is rarely feasible. In EBV-seronegative children, infection most commonly develops within the first year after transplantation, particularly between 3 and 6 months post-transplant, coinciding with the period of highest immunosuppressive intensity [18]. Consequently, identifying an effective prophylaxis in the first months after transplantation for children with an EBV D+/R− mismatch is crucial to prevent infection and reduce the risk of PTLD.

A recent large European survey summarized the current practices for EBV monitoring and prevention strategies in pediatric solid organ transplantation. All participating centers reported regular monitoring of EBV viral load in peripheral blood to identify patients at risk, while a growing proportion also incorporates adjunctive immune assays to improve predictive accuracy. However, the lack of standardized viral load thresholds and inter-laboratory variability remain major limitations to defining early intervention criteria. Most centers still rely on pre-emptive reduction in immunosuppression as the main preventive measure, whereas the use of antivirals is uncommon due to limited supporting evidence [3]. Therefore, no standardized or universally effective preventive strategy against primary EBV infection has yet been established and current approaches vary considerably among transplant centers.

Our study highlights that, although IVIG has been proposed as a possible strategy for the prophylaxis of EBV infection, it does not appear to be an effective prevention strategy in this high-risk population of kidney-transplanted children. IVIG treatment failed to prevent primary EBV infection, and our findings are consistent with a randomized trial in pediatric liver transplant recipients using high-titer CMV-IVIG, which showed no significant reduction in EBV disease or PTLD, although a trend toward lower incidence was observed [8].

Furthermore, our results indicated a possible increased risk of infection in treated patients (HR 3.24, 95% CI 0.87–12.01; *p* = 0.079), as children in the treatment group had a higher incidence of EBV-DNA positivity during follow-up compared to the control group (64% vs. 25%, *p* = 0.049). In the treatment group, the prevalence of EBV infection was already 57% at 6 months, remained constant during the first two years, and then decreased to 36% at 5 years, whereas in the control group prevalence remained stable at 20–25%. Although the multivariable Cox regression model for EBV-DNA positivization reached overall statistical significance, no single covariate emerged as an independent predictor. This apparent inconsistency may be largely explained by the limited sample size, which reduces statistical power and complicates the interpretation of multivariable effects. Notably, although age was the only variable significantly different between the study groups, it did not emerge as a predictor of EBV-DNA positivization in the Cox model. This finding suggests that the observed intergroup difference in age likely reflects baseline heterogeneity rather than a true independent risk factor. The modest number of events and the potential collinearity among clinical variables may further account for the lack of robust associations.

The higher incidence of primary EBV infection observed in the IVIG group may be explained by several factors. Differences in age distribution between groups might influence the likelihood of primary EBV exposure; however, since EBV circulation peaks during adolescence, a higher infection rate would be expected in the older control group [19]. Intrafamilial or interpersonal transmission was also evaluated, yet both groups adhered to identical post-transplant infection-prevention measures, avoiding school and social activities in the early post-transplant period, making this hypothesis unlikely. Furthermore, differences in immunosuppressive management could theoretically affect viral susceptibility, but this appears improbable given the comparable incidence of rejection episodes and similar calcineurin inhibitor trough levels observed between groups.

One possible explanation for the higher incidence of EBV infection observed in the IVIG group is the theoretical possibility of residual viral transmission through immunoglobulin preparations. In a review addressing the problem of viral infections transmitted through blood transfusions, the authors focused on the main viruses involved, their clinical impact, and preventive strategies, concluding that, although transfusion-associated viral infections cannot be completely eliminated, their incidence can be significantly reduced through appropriate preventive measures [20].

Although the fractionation and solvent–detergent steps employed in modern manufacturing remove or inactivate most enveloped viruses, EBV persists latently within memory B-cells; cell-free antibody preparations cannot contain intact lymphocytes, but residual viral DNA or small extracellular vesicles could theoretically seed infection in naïve hosts. Evidence that blood products can transmit EBV, even after universal leukoreduction, is growing. In a Canadian pediatric stem-cell cohort (TREASuRE study) a single geno-matched transmission was documented after 87 red-cell and 238 platelet units, illustrating that the event is rare but possible in the setting of profound immunosuppression [21]. A previous retrospective cohort from the same group demonstrated a positive, dose-dependent association between transfusion volume and post-transplant EBV infection; recipients in the highest platelet-exposure tertile (>2.5 L) had a two-fold risk increase (RR 2.19, 95% CI 1.21–3.97) [22]. In our cohort, the two study groups received a comparable number of blood product transfusions in the immediate post-transplant period (IVIG group 57%, control group 50%, *p* = 0.732), suggesting that the higher incidence of EBV infection observed in the IVIG group may be related to the repeated administration of immunoglobulin prophylaxis rather than differences in transfusion exposure. Conversely, two adolescent EBV-naïve recipients deliberately transfused with small volumes of blood from their living EBV-positive donors developed asymptomatic seroconversion before transplant and remained PTLD-free for five years [23]. While obviously not generalizable, these observations underscore the biological plausibility that B-cell-containing blood products (or derivatives prepared from large donor pools) can transmit infectious EBV.

Taken together, our findings seem to suggest caution in prescribing repeated IVIG infusions as EBV prophylaxis in pediatric kidney recipients. However, while residual viral transmission through immunoglobulin products is theoretically possible, current evidence does not support a causal relationship between IVIG administration and EBV infection. Therefore, this hypothesis should therefore be interpreted with caution and regarded as speculative.

Other drugs that have been studied for EBV prophylaxis in children include valganciclovir and rituximab. In our cohort, 73% of patients received valganciclovir for CMV prophylaxis, with similar use in the IVIG and control groups (IVIG group 77%, control group 69%, *p* = 0.523), confirming that antiviral prophylaxis with valganciclovir has not proven to prevent EBV replication or PTLD [24,25,26,27].

The monoclonal anti-CD20 antibody rituximab has been successfully used to achieve viral clearance in cases of persistent EBV-DNA or early PTLD, particularly in hematopoietic stem cell transplant recipients and after solid organ transplantation [28,29,30,31]. In our cohort, only two patients received rituximab, which limits any conclusions regarding its impact within the present study.

In summary, IVIG prophylaxis failed to prevent primary EBV infection or enhance long-term immunity, suggesting that its routine use in pediatric D+/R− kidney recipients is not justified. Larger prospective studies are needed to confirm these results.

### Limitations

This study has several limitations that should be acknowledged. First, the small sample size limits the statistical power and precludes more detailed subgroup analyses, particularly regarding age stratification, immunosuppressive regimens, or virological kinetics. Second, the retrospective and non-randomized design may have introduced selection and information biases, despite the homogeneous inclusion criteria and standardized institutional follow-up. Third, a significant baseline age difference was observed between groups, which could have influenced viral exposure risk or immune maturation, although all patients were EBV-seronegative at the time of transplantation. Finally, this single-center experience may not be fully generalizable to other centers with different transplant protocols or patient populations. Nonetheless, our findings provide the first pediatric data exploring IVIG prophylaxis for EBV in kidney transplantation and may serve as a basis for future multicenter prospective studies.

## 5. Conclusions

Our data suggest that IVIG infusions do not appear to be as effective as prophylaxis for primary EBV infection in high-risk pediatric kidney transplant recipients with an EBV D+/R− mismatch. However, our study has several limitations, including its retrospective design and the small number of patients enrolled. A potential selection bias is related to older age at transplantation in the control group; nevertheless, given the negative EBV serology at the time of transplant, this difference was not considered relevant for interpreting the results. Despite these limitations, this study represents the first documented pediatric experience evaluating the use of IVIG as prophylaxis against EBV in kidney transplantation, providing useful preliminary data in an area where evidence is currently lacking. Further studies are warranted to confirm these findings.

## Figures and Tables

**Figure 1 medicina-61-01967-f001:**
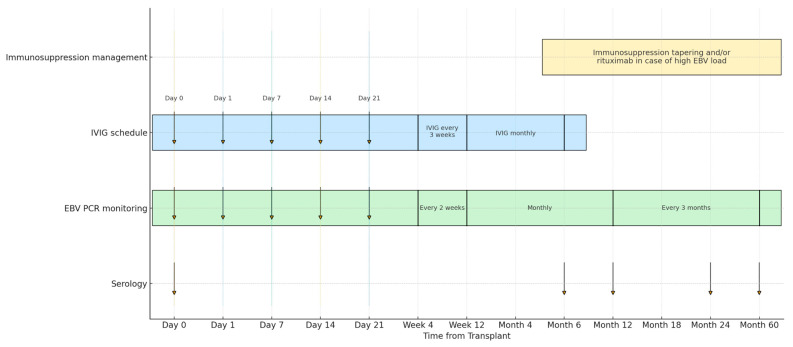
Procedural Timeline: IVIG Prophylaxis, EBV PCR, Serology, and Immunosuppression From Transplant to 60 Months.

**Figure 2 medicina-61-01967-f002:**
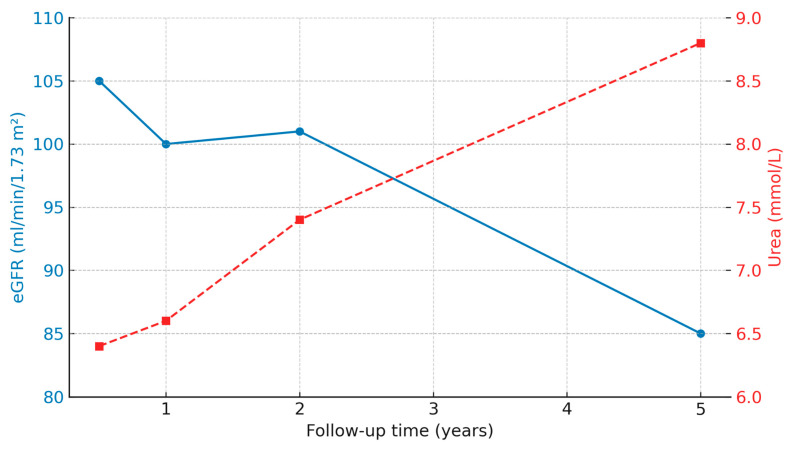
Longitudinal trend of eGFR and Urea in the population. Friedman test: eGFR X7(3) = 19.2, *p* < 0.001 | Urea X7(3) = 12.7, *p* = 0.005.

**Figure 3 medicina-61-01967-f003:**
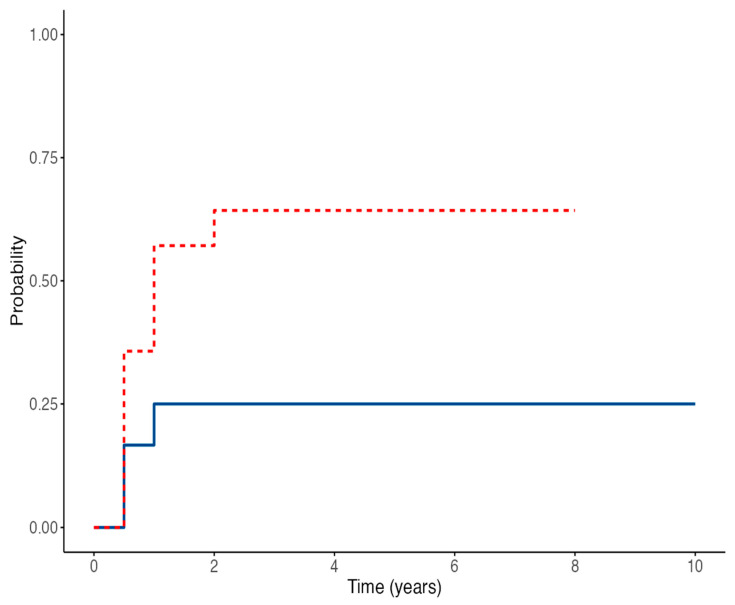
Cumulative EBV-DNA positivity in the IVIG-treated group (red line) and control group (blue line) over time (*p* = 0.079).

**Figure 4 medicina-61-01967-f004:**
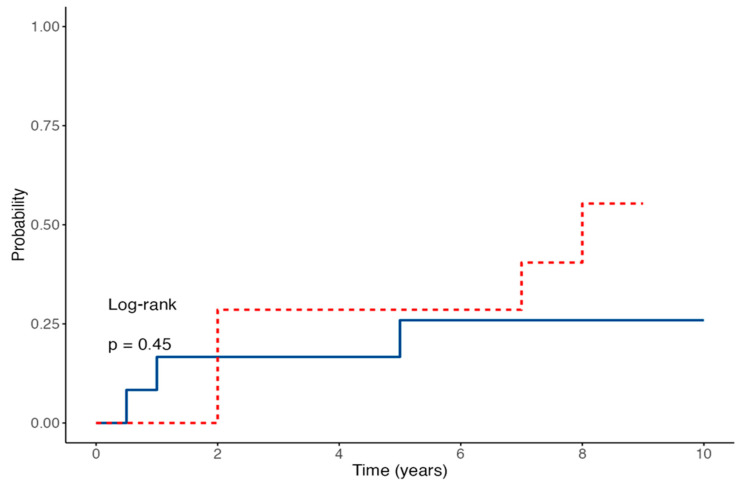
Cumulative anti-EBNA IgG seroconversion in the IVIG-treated group (red line) and control group (blue line) over time (*p* = 0.45).

**Table 1 medicina-61-01967-t001:** Baseline Characteristics of IVIG and control groups.

Variable	IVIG Group (*n* = 14)	Control Group (*n* = 12)	*p* Value
Age at transplant (years) (range)	4.1 (2.3–7.0)	8.2 (7.0–12.1)	0.008
Sex (male) (%)	9 (64)	8 (67)	0.919
Weight at transplant (kg) (range)	10.9 (5.7–28.5)	21.0 (19.4–44.0)	0.084
Number of second transplants (n) (%)	0 (0)	3 (25)	0.098
Mismatch (n) (range)	4 (2–5)	3 (2–4)	0.255
Living donor transplant (n) (%)	4 (29)	4 (33)	0.796
ATG * (n) (%)	1 (7)	3 (25)	0.272
Tacrolimus (n) (%)	7 (50)	6 (50)	1.000
PTLD * (n) (%)	1 (7)	0 (0)	0.619
Valganciclovir (n) (%)	11 (79)	8 (67)	0.523
Rituximab (n) (%)	1 (7)	0 (0)	0.619
Transfusion of blood, platelet or plasma (n) (%)	8 (57)	6 (50)	0.732
Pre-emptive transplant (n) (%)	1 (7)	5 (42)	0.070
Peritoneal dialysis (n) (%)	9 (64)	6 (50)	0.460
Hemodialysis (n) (%)	0 (0)	1 (8)	0.471
Hemodialysis and peritoneal dialysis (n) (%)	4 (29)	0 (0)	0.080
CAKUT * (n) (%)	9 (64)	7 (58)	0.826
Glomerulopathy (n) (%)	1 (7)	3 (25)	0.218
Ciliopathy (n) (%)	3 (21)	1 (8)	0.381
Asphyxia (n) (%)	1 (7)	1 (8)	1.000

* ATG, Anti-thymocyte globulin; PTLD, Post-transplant lymphoproliferative disorder; CAKUT, congenital anomalies of the kidney and the urinary tract.

**Table 2 medicina-61-01967-t002:** Renal function and immunosuppressive parameters over time according to study group (Control vs. IVIG).

Parameter	Time Point	N	Control Group Median [IQR]	IVIG Group Median [IQR]	*p*-Value
eGFR (mL/min/1.73 m^2^)	6 months	26	104.8 [84.9–118.0]	108.0 [102.0–118.5]	0.35
	12 months	25	97.8 [89.1–122.3]	106.0 [100.0–131.3]	0.24
	24 months	25	95.8 [70.7–117.3]	102.5 [92.0–114.2]	0.33
	60 months	23	50.8 [48.3–94.8]	84.5 [63.0–97.9]	1.00
Urea (mmol/L)	6 months	26	4.9 [4.8–7.6]	6.2 [6.2–6.6]	0.42
	12 months	25	5.2 [5.5–8.2]	6.6 [6.6–8.4]	0.94
	24 months	26	5.3 [5.7–9.8]	7.5 [6.9–9.1]	0.84
	60 months	23	6.2 [7.0–12.3]	9.2 [8.1–11.7]	0.72
Uric acid (mmol/L)	6 months	26	0.3 [0.3–0.3]	0.3 [0.3–0.4]	0.96
	12 months	25	0.2 [0.2–0.4]	0.3 [0.3–0.4]	0.79
	24 months	26	0.2 [0.3–0.4]	0.3 [0.3–0.4]	0.19
	60 months	23	0.3 [0.3–0.3]	0.3 [0.3–0.4]	0.40
Tacrolimus (ng/mL)	6 months	10	5.0 [6.3–9.0]	7.2 [7.8–8.0]	0.75
	12 months	13	2.3 [5.6–7.0]	5.3 [6.7–7.6]	0.58
	24 months	19	4.3 [5.3–6.3]	6.2 [6.1–6.9]	0.94
	60 months	17	5.3 [4.9–7.1]	6.8 [6.5–7.2]	0.68
Cyclosporine (C2, ng/mL)	6 months	16	524.8 [512.7–748.0]	647.0 [674.0–825.2]	0.87
	12 months	12	314.6 [579.9–708.0]	579.5 [653.0–739.2]	0.63
	24 months	7	513.3 [420.0–626.0]	730.0 [523.0–781.3]	0.28
	60 months	5	398.2 [577.0–714.0]	494.0 [645.5–879.0]	0.64

Abbreviations: eGFR = estimated glomerular filtration rate; IVIG = intravenous immunoglobulin; C2 = cyclosporine 2 h post-dose level; N = number of non-missing values. All values are presented as median [IQR].

**Table 3 medicina-61-01967-t003:** Virological and serological data at different timepoints in IVIG and control groups.

Category	Time-Point	IVIG Group	Control Group	*p*-Value
EBV-DNA	6 months	8/14 (57%)	3/12 (25%)	0.11
	12 months	8/14 (57%)	3/12 (25%)	0.11
	24 months	7/14 (50%)	3/12 (25%)	0.22
	60 months	4/14 (28%)	2/12 (17%)	0.38
	Ever positive	9/14 (64%)	3/12 (25%)	0.047 *
Anti-EBNA IgG	6 months	9/14 (64%)	2/12 (17%)	0.012 *
	12 months	0/14 (0%)	2/12 (17%)	0.23
	24 months	3/14 (21%)	3/12 (25%)	0.83
	60 months	5/14 (36%)	2/12 (17%)	0.15
	Ever positive	6/14 (43%)	3/12 (25%)	0.38

* indicates statistical significance (*p* < 0.05).

**Table 4 medicina-61-01967-t004:** Cox regression analysis of factors associated with EBV DNA positivization.

Variable	Category	n (%)	HR (Univariable)	HR (Multivariable)
Outcome (Case/Control)	0 (Control)	12 (46.2)	–	–
	1 (Case)	14 (53.8)	3.24 (0.87–12.01), *p* = 0.079	1.50 (0.33–6.73), *p* = 0.599
Tacrolimus	0 (No)	13 (50.0)	–	–
	1 (Yes)	13 (50.0)	0.44 (0.13–1.46), *p* = 0.181	0.30 (0.08–1.14), *p* = 0.078
Primary cause of CKD	CAKUT	16 (61.5)	–	–
	Asphyxia	2 (7.7)	1.21 (0.15–9.69), *p* = 0.857	1.88 (0.19–18.41), *p* = 0.587
	Ciliopathy	4 (15.4)	2.05 (0.54–7.77), *p* = 0.292	3.65 (0.81–16.57), *p* = 0.093
	Glomerulopathy	4 (15.4)	0.00 (0.00–Inf), *p* = 0.998	0.00 (0.00–Inf), *p* = 0.998
Age at transplant (years)	Mean (SD)	6.9 (4.9)	0.88 (0.75–1.04), *p* = 0.131	0.90 (0.75–1.07), *p* = 0.238

## Data Availability

The datasets are not publicly available due to privacy and ethical restrictions. However, they are available from the corresponding author on reasonable request and with permission from the Ethics Committee, if applicable.

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
