# Peer review of "Are Intravenous Immunoglobulins Effective in Preventing Primary EBV Infection in Pediatric Kidney Transplant Recipients?"

_medicina, 2025, doi:10.3390/medicina61111967_

Round 1

Reviewer 1 Report

Comments and Suggestions for Authors

The study by Partigiani et al. evaluated the effect of immunoglobulin on the prevention of Epstein–Barr virus (EBV) infection in a high-risk pediatric kidney transplant population with an EBV D+/R− mismatch (n = 26). Although the results did not demonstrate a clear beneficial effect, please address the following points before we proceed:

  1. Were there any differences in the rate of rejection or in the use of immunosuppressive therapy to manage rejection episodes between groups?

  2. Similarly, were the trough levels of calcineurin inhibitors different across groups, potentially indicating variations in the degree of immunosuppression?

Author Response

We sincerely thank the reviewers for their thoughtful and constructive comments. Here we provide a point-by-point response to all the reviewers’ comments.

Comment 1:  Were there any differences in the rate of rejection or in the use of immunosuppressive therapy to manage rejection episodes between groups?

Response 1: We thank the reviewer for this comment. The incidence of biopsy-proven rejection within the first five years after transplantation was comparable between the two groups (57% in the IVIG group vs. 58% in the control group; p = 0.838, log-rank test). It has been included in the results section (page 7, lines 275-277). Moreover, both groups received the same standardized immunosuppressive regimen and rejection management protocol. This definition and treatment approach have been clarified in the Methods section (see Methods, page 3, lines 135-145, highlighted in green).

Comment 2: Similarly, were the trough levels of calcineurin inhibitors different across groups, potentially indicating variations in the degree of immunosuppression?

Response 2: We thank the reviewer for this observation. Trough levels of calcineurin inhibitors (tacrolimus and cyclosporine C2) have been included and reported in the Results (“Kidney function and laboratory tests over time”). No significant differences were observed between the IVIG and control groups at any follow-up time point, indicating a comparable degree of immunosuppression throughout the study period (Results, Section 3.2, page 6-7, highlighted in green).

Reviewer 2 Report

Comments and Suggestions for Authors

I appreciate the opportunity to review this manuscript. The study addresses a highly relevant clinical topic in pediatric nephrology and kidney transplantation: the prevention of primary Epstein–Barr virus (EBV) infection and the reduction of post-transplant lymphoproliferative disorder (PTLD) risk. The topic is pertinent, well contextualized, and responds to an unresolved clinical need.

  1. General Assessment

The manuscript is clearly structured, fluently written, and methodologically sound for a retrospective study. The authors present the objectives, study population, and results in an organized manner.

Main strengths include:

  • The long follow-up period (median 7.5 years), which provides clinical depth and temporal consistency.
  • The balanced discussion, in which the authors acknowledge the study’s limitations without overinterpreting the findings.

Areas for improvement:

  • The small sample size (n = 26) limits statistical power.
  • The single-center, retrospective design restricts generalizability.
  • The age difference between groups could introduce potential confounding bias.
  • The interpretation of possible viral transmission through IVIG should be presented more cautiously and accompanied by a note of scientific prudence.
  1. Section-by-Section Comments

Title and Abstract

  • It is recommended to add the phrase “retrospective case-control study” for greater precision.
  • The abstract is clearly written; however, it should specify from the first line that the study is retrospective.
  • Suggest rephrasing “IVIG prophylaxis was unexpectedly associated with a higher cumulative incidence” for smoother academic English (e.g., “IVIG prophylaxis was unexpectedly linked to a higher cumulative incidence of EBV infection”).

Introduction

  • A concluding paragraph should be added to state the explicit objective and research hypothesis of the study.

Materials and Methods

  • The section could be strengthened by clarifying inclusion and exclusion criteria and stating whether any patients were lost to follow-up.
  • Mentioning the use of Jamovi and R is appropriate, though it could be abbreviated with a concise citation of version and license.

Results

  • Results are well organized and supported by clear tables.
  • There is consistency between data and conclusions.
  • The interpretation of non-significant differences (p > 0.05) is correct and cautious.
  • Consider improving repetitive phrasing such as “EBV-DNA was positive in…” by syntactic unification or using a summarized tabular format.

Discussion

  • The hypothesis regarding potential viral transmission through IVIG is interesting but should be presented in more speculative and less assertive terms, for example:

“Although residual viral transmission through immunoglobulin preparations is theoretically possible, current evidence does not support a causal relationship.”

  • The integration of updated and relevant references is commendable.
  • Grammar and punctuation are correct; however, some long paragraphs could be divided to enhance readability.

Conclusions

  • It would be useful to emphasize the scientific contribution, for example:

“This study represents the first documented pediatric experience evaluating the use of IVIG as prophylaxis against EBV in kidney transplantation.”

  1. Minor Recommendations
  • Standardize the format of abbreviations throughout the text (e.g., define EBV-DNA only at first mention).
  • Review the use of capitalization in virus and antigen names (EBV, EBNA-IgG, etc.).
  • Simplify some sentences to improve fluency (e.g., replace redundant passive constructions).
  • If possible, include 3–5 clinical take-home messages in the conclusions or as a summary box:
    1. IVIG prophylaxis does not reduce primary EBV infection.
    2. It does not confer sustained immunological advantage.
    3. Routine use in pediatric D+/R− kidney recipients is not justified.
    4. Larger, prospective studies are required to confirm these findings.

Author Response

We sincerely thank the reviewers for their thoughtful and constructive comments, and especially for their positive remarks and for acknowledging the significance of our study. Here we provide a point-by-point response to all the reviewers’ comments.

Title and Abstract

Comment 1: It is recommended to add the phrase “retrospective case-control study” for greater precision. The abstract is clearly written; however, it should specify from the first line that the study is retrospective. Suggest rephrasing “IVIG prophylaxis was unexpectedly associated with a higher cumulative incidence” for smoother academic English (e.g., “IVIG prophylaxis was unexpectedly linked to a higher cumulative incidence of EBV infection”).

Response 1: We have made the suggested changes to the abstract (see Abstract, page 1, lines 17 and 28, highlighted in green).

Comment 2:  Introduction:  A concluding paragraph should be added to state the explicit objective and research hypothesis of the study .

Response 2: We thank the reviewer for this helpful suggestion. We have revised the final paragraph of the Introduction to clearly state the underlying research hypothesis and the explicit objective of the study (see Introduction, page 2, lines 87-93, highlighted in green),

Materials and Methods:

Comment 3: 

  1. The section could be strengthened by clarifying inclusion and exclusion criteria and stating whether any patients were lost to follow-up
  2. Mentioning the use of Jamovi and R is appropriate, though it could be abbreviated with a concise citation of version and license.

Response 3:

  1. a) We thank the reviewer for this helpful suggestion. We have now specified the inclusion and exclusion criteria and clarified the follow-up completeness (see Methods, page 3, lines 103-113, highlighted in green). All patients reached the 24-month follow-up; three were transferred to other centers after two years, while the remaining participants completed at least five years of observation.
  2. b) We abbreviated the sentence as suggested (see Methods, page 5, lines 221.223, highlighted in green).

Results

Comment 4

Consider improving repetitive phrasing such as “EBV-DNA was positive in…” by syntactic unification or using a summarized tabular format.

Response 4: We thank the reviewer for this suggestion. We have revised the paragraph 3.2 (now 3.3) of the Results describing EBV-DNA positivity to avoid repetitive phrasing (see Results, paragraph 3.3, page 8, lines 294-299, highlighted in green).

Discussion

Comment 5

  1. The hypothesis regarding potential viral transmission through IVIG is interesting but should be presented in more speculative and less assertive terms, for example:

“Although residual viral transmission through immunoglobulin preparations is theoretically possible, current evidence does not support a causal relationship.”

  1. Grammar and punctuation are correct; however, some long paragraphs could be divided to enhance readability.

Response 5:

  1. We agree that the hypothesis regarding potential viral transmission through IVIG should be presented in more speculative and less assertive terms. We have revised the introductory and concluding sentences of the relevant paragraph in the Discussion to adopt a more speculative tone (see Discussion, page 11, lines 423-425 and lines 453-458, highlighted in green)
  2. We thank the reviewer for this helpful suggestion. We have divided the longest paragraphs of the Discussion into shorter paragraphs to improve readability (see Discussion section).

Conclusions

Comment 6

It would be useful to emphasize the scientific contribution, for example:

“This study represents the first documented pediatric experience evaluating the use of IVIG as prophylaxis against EBV in kidney transplantation.”

Response 6: We thank the reviewer for this valuable suggestion. We have revised the Conclusion to better highlight the scientific contribution of our study (see Conclusions, page 14, lines 497-499)

Minor Recommendations

Comment 7:

  1. Standardize the format of abbreviations throughout the text (e.g., define EBV-DNA only at first mention).
  2. b) Review the use of capitalization in virus and antigen names (EBV, EBNA-IgG, etc.).
  3. c) Simplify some sentences to improve fluency (e.g., replace redundant passive constructions).
  4. d) If possible, include 3–5 clinical take-home messages in the conclusions or as a summary box:
    • IVIG prophylaxis does not reduce primary EBV infection.
    • It does not confer sustained immunological advantage.
    • Routine use in pediatric D+/R− kidney recipients is not justified.
    • Larger, prospective studies are required to confirm these findings.

Response 7:

  1. We have checked the entire manuscript and corrected abbreviations where necessary.
  2. We have checked the entire manuscript and corrected capitalization where necessary
  3. We have revised the manuscript to simplify several sentences and replace redundant passive constructions (see Introduction, page 1, lines 66-67, highlighted in sky blue; Methods, paragraph 2.1 Study Design and Subjects, page 3, line 115, and page 4, line 157, highlighted in sky blue; Results, paragraph 3.1 Population characteristics, page 6, lines 234-236, highlighted in sky blue).
  4. We thank the reviewer for this suggestion. We have added a concise summary sentence at the end of the Discussion to avoid redundancy with the Conclusions section (see end of the Discussion section, page 13, lines 469-471, highlighted in green).

Reviewer 3 Report

Comments and Suggestions for Authors

The topic is significant, important and always innovative!

Some important things are missing that need to be corrected and included in the second version:

- It is necessary to state the Institution's Ethical Consent (date and decision number) with regard to the topic and therapy!
- Define the broad spectrum of lymphoproliferative disease and the current standard therapy!

- Kidney function is missing in the tables!
It is necessary to specify the values ​​of creatinine, urea, urate, Tacrolimus level!
- In patients treated with anti CD 20 antibody
is it necessary to state whether the CD 20 cells are determinable?
- Creatinine clearance and proteinuria must be listed in the table at the beginning and end of therapy!

- A wider discussion must be included! More references for interpretation are missing for this well-known topic!

Author Response

We sincerely thank the reviewers for their thoughtful and constructive comments, and especially for their positive remarks and for acknowledging the significance of our study. Here we provide a point-by-point response to all the reviewers’ comments.

Comment 1: It is necessary to state the Institution's Ethical Consent (date and decision number) with regard to the topic and therapy!

Response 1: We thank the reviewer for this observation. The ethical approval and consent statement are already reported in the manuscript, under the Institutional Review Board Statement: approval by the Ethics Committee of Padua University Hospital (protocol code 65023/2021, approval date 23 September 2021), page 14, lines 507-510.

Comment 2: Define the broad spectrum of lymphoproliferative disease and the current standard therapy!

Response 2: We thank the reviewer for this helpful comment. We have expanded the description of PTLD and the current standard therapeutic approach in the Introduction (see Introduction, page 2, lines 52-57, highlighted in green).

Comment 3: Kidney function is missing in the tables! It is necessary to specify the values ​​of creatinine, urea, urate, Tacrolimus level!

Response 3: We thank the reviewer for this valuable comment. In the revised version, kidney function parameters have been comprehensively included and analyzed. Specifically, serum creatinine, urea, uric acid, tacrolimus, and cyclosporine (C2) levels were evaluated longitudinally at 6, 12, 24, and 60 months post-transplant and are now reported in the new subsection (see Results, paragraph “3.2 Kidney function and laboratory tests over time”, Figure 2, Table 2, highlighted in green). 

Comment 4:  In patients treated with anti CD 20 antibody is it necessary to state whether the CD 20 cells are determinable?

Response 4: We thank the reviewer for this valuable comment. Only one patient in our cohort received rituximab, and CD20 cell depletion was confirmed following treatment. The detailed pre-infusion and post-treatment assessment, including CD20 monitoring, has been described in the Methods section (see Methods, page 4, lines 166-174, highlighted in green).

Comment 5: Creatinine clearance and proteinuria must be listed in the table at the beginning and end of therapy!

Response 5: We thank the reviewer for this comment. Baseline and 6-month creatinine clearance (eGFR) and proteinuria (UPCR) were evaluated during the analysis. Median eGFR values were 106 mL/min/1.73 m² at baseline and 101 mL/min/1.73 m² at 6 months, while median UPCR at 6 months was 0.23 mg/mg. No significant differences were observed between the IVIG and control groups for either parameter. Given the absence of statistically or clinically relevant findings—and considering that these variables were not primary study outcomes—we preferred not to include them in the main Results section. Moreover, UPCR was not assessed in the immediate post-transplant period, as protein excretion is transiently altered at this stage and would not accurately reflect graft function. It should also be noted that IVIG therapy was initiated at the transplantation, as described in the Methods section (see Methods, page 3, lines 121-125, highlighted in green).

Comment 6: A wider discussion must be included! More references for interpretation are missing for this well-known topic!

Response 6: We thank the reviewer for this valuable suggestion. We have expanded the Discussion accordingly and added relevant references (see Discussion, page 11-12, lines 360-387 and References, highlighted in green).

Reviewer 4 Report

Comments and Suggestions for Authors

The article “Are Intravenous Immunoglobulins Effective in Preventing Primary EBV Infection in Pediatric Kidney Transplant Recipients?” is a well-written and clinically relevant single-center case-control study addressing an important question in pediatric kidney transplantation. The manuscript is generally clear and the methodology is sound for a retrospective study. The findings are unexpected, but very noteworthy.

This study provides valuable evidence suggesting that IVIG prophylaxis is not effective and may potentially be harmful in preventing primary EBV infection in high-risk pediatric kidney transplant recipients. The long follow-up period is a significant strength, but the study's limitations, primarily its retrospective design and small sample size, require a caution when drawing conclusions.

The study addresses a clear clinical dilemma in a well-defined, high-risk population. A median follow-up of 7.5 years provides robust data on long-term outcomes. The primary and secondary endpoints are well-defined and clinically relevant. The protocol for EBV DNA monitoring is comprehensive and follows standard clinical practice.

There are, however, some suggestions for improvement.

In the Results sub-section of the Abstract, the authors could consider explicitly mentioning the significant age difference between groups.

The Introduction section effectively sets the stage by explaining the clinical problem, the high-risk population, and the theoretical rationale for using IVIG. The authors could briefly mention the existing conflicting or limited evidence on IVIG for EBV prophylaxis in solid organ transplantation to better justify the need for this study.

Regarding the Materials and Methods section, the sub-section 2.1 could be more clearly explained. Perhaps a summary table or a consolidated timeline in the text would be helpful.

In the same section, the significant age difference between groups is a critical potential confounder since younger age is itself a risk factor for primary EBV infection and potentially different immune responses. The current univariate analysis could seem insufficient. Therefore, the statistical analysis could even be more strengthened by performing a multivariate Cox regression analysis that includes age and any other imbalanced baseline characteristic as covariates. This would help determine if the association between IVIG and higher EBV infection risk is independent of the younger age of the treatment group.

For the Results section, no significant improvements are necessary.

The Discussion section excellently contextualizes the findings within existing literature and thoughtfully explores potential mechanisms for the unexpected result.

The authors should consider starting the discussion by directly stating the main, counter-intuitive result that IVIG was associated with a higher (not lower!) incidence of EBV infection. Other possible immunological mechanisms for this, beyond potential viral transmission, should also be discussed.

The limitations should be expanded into a dedicated paragraph at the end of the Discussion section. Key limitations are small sample size (low statistical power, inability to perform robust subgroup analyses), retrospective, non-randomized design, significant baseline age difference, and single-center experience (which may limit generalizability).

The Conclusion section is clear.

Implementing the mentioned suggestions will enhance the study’s rigor, clarity, and impact, and besides these, no further corrections are needed.

Author Response

We sincerely thank the reviewers for their thoughtful and constructive comments, and especially for their positive remarks and for acknowledging the significance of our study. Here we provide a point-by-point response to all the reviewers’ comments.

Comment 1: In the Results sub-section of the Abstract, the authors could consider explicitly mentioning the significant age difference between groups.

Response 1: We thank the reviewer for this useful suggestion. The significant age difference between groups has now been explicitly mentioned in the Results subsection of the Abstract, including the corresponding data (see Abstract, page 1, lines 25-28, highlighted in green).

Comment 2: Introduction section -  The authors could briefly mention the existing conflicting or limited evidence on IVIG for EBV prophylaxis in solid organ transplantation to better justify the need for this study.

Response 2: We appreciate the suggestion. We have amended the Introduction to briefly summarize the state of the evidence (page 2, lines 77-82).

Comment 3: Regarding the Materials and Methods section, the sub-section 2.1 could be more clearly explained. Perhaps a summary table or a consolidated timeline in the text would be helpful.

Response 3:  We thank the reviewer for this helpful suggestion. We have revised Section 2.1 (Study Design and Subjects) to improve clarity and flow by explicitly describing the study setting, inclusion criteria, group allocation (IVIG vs. no prophylaxis) with calendar windows, the IVIG dosing schedule, and the EBV surveillance protocol. In addition, we added a consolidated timeline (now Figure 1: “Post-Transplant Timeline of IVIG Prophylaxis, EBV Monitoring, Serology, and Immunosuppression Management (Months 0–60)”) .

Comment 4: In the same section, the significant age difference between groups is a critical potential confounder since younger age is itself a risk factor for primary EBV infection and potentially different immune responses. The current univariate analysis could seem insufficient. Therefore, the statistical analysis could even be more strengthened by performing a multivariate Cox regression analysis that includes age and any other imbalanced baseline characteristic as covariates. This would help determine if the association between IVIG and higher EBV infection risk is independent of the younger age of the treatment group.

Response 4: Thank you for your valuable comment. We agree that age is a potential confounder, as younger patients are more susceptible to primary EBV infection. Accordingly, we performed a multivariable Cox regression including age and other baseline imbalances. Although the overall model was significant, age did not emerge as an independent predictor, suggesting that the association between IVIG and EBV DNA positivization is not solely age-related. See methods (Page 5, lines 216-220), results 3.2 (Page 7), discussion (page 12, lines 401-410)

Comment 5: Discussion section - The authors should consider starting the discussion by directly stating the main, counter-intuitive result that IVIG was associated with a higher (not lower!) incidence of EBV infection.

Response 5: We thank the reviewer for this insightful comment. We have revised the beginning of the Discussion to immediately emphasize the main, counter-intuitive finding of our study (see Discussion, page 11, lines 352–356, highlighted in green).

Comment 6: Discussion - Other possible immunological mechanisms for this, beyond potential viral transmission, should also be discussed.

Response 6: We thank the reviewer for this comment. We have now expanded the Discussion to include additional considerations that may explain the higher incidence of primary EBV infection observed in the IVIG group (see Discussion, page 12, lines 411–425). We have also added details on our infection-prevention protocol in the Methods section (see Methods, page 3, lines 131.134, highlighted in green).

Comment 7: The limitations should be expanded into a dedicated paragraph at the end of the Discussion section. Key limitations are small sample size (low statistical power, inability to perform robust subgroup analyses), retrospective, non-randomized design, significant baseline age difference, and single-center experience (which may limit generalizability).

Response 7: We sincerely thank the reviewer for this constructive comment. In accordance with the suggestion, we have expanded the end of the Discussion by adding a dedicated paragraph on the study limitations. We believe that this addition improves the transparency and interpretability of our findings (see Discussion, page 9, lines 474-485).

Round 2

Reviewer 3 Report

Comments and Suggestions for Authors

No